# Genome-wide association mapping reveals race-specific SNP markers associated with anthracnose resistance in carioca common beans

**Caléo Panhoca de Almeida**[1]⊙*, **Jean Fausto de Carvalho Paulino**[1]⊙, **Caio Cesar Ferrari Barbosa**[1], **Gabriel de Moraes Cunha Gonçalves**[2], **Roberto Fritsche-Neto**[3], **Sérgio Augusto Morais Carbonell**[2], **Alisson Fernando Chiorato**[2], **Luciana Lasry Benchimol-Reis**[1]

1 Centro de Pesquisa em Recursos Genéticos Vegetais, Instituto Agronômico (IAC), Campinas, São Paulo, Brazil, 2 Centro de Grãos e Fibras do Instituto Agronômico (IAC), Campinas, São Paulo, Brazil, 3 Departamento de Genética, Universidade de São Paulo (ESALQ-USP), Piracicaba, São Paulo, Brazil

⊙ These authors contributed equally to this work.
* caleoalmeida@hotmail.com

**Data Availability Statement:** All relevant data are within the manuscript and its Supporting Information files.

## Abstract

Brazil is the largest consumer of dry edible beans (*Phaseolus vulgaris* L.) in the world, 70% of consumption is of the carioca variety. Although the variety has high yield, it is susceptible to several diseases, among them, anthracnose (ANT) can lead to losses of up to 100% of production. The most effective strategy to overcome ANT, a disease caused by the fungus *Colletotrichum lindemuthianum*, is the development of resistant cultivars. For that reason, the selection of carioca genotypes resistant to multiple ANT races and the identification of *loci*/markers associated with genetic resistance are extremely important for the genetic breeding process. Using a carioca diversity panel (CDP) with 125 genotypes and genotyped by BeadChip BARCBean6K_3 and a carioca segregating population AM (AND-277 × IAC-Milênio) genotyped by sequencing (GBS). Multiple interval mapping (MIM) and genome-wide association studies (GWAS) were used as mapping tools for the resistance genes to the major ANT physiological races present in the country. In general, 14 single nucleotide polymorphisms (SNPs) showed high significance for resistance by GWAS, and *loci* associated with multiple races were also identified, as the *Co-3 locus*. The SNPs ss715642306 and ss715649427 in linkage disequilibrium (LD) at the beginning of chromosome Pv04 were associated with all the races used, and 16 genes known to be related to plant immunity were identified in this region. Using the resistant cultivars and the markers associated with significant quantitative resistance *loci* (QRL), discriminant analysis of principal components (DAPC) was performed considering the allelic contribution to resistance. Through the DAPC clustering, cultivar sources with high potential for durable anthracnose resistance were recommended. The MIM confirmed the presence of the *Co-1⁴ locus* in the AND-277 cultivar which revealed that it was the only one associated with resistance to ANT race 81. Three other *loci* were associated with race 81 on chromosomes Pv03, Pv10, and Pv11. This is the first study to identify new resistance *loci* in the AND-277 cultivar. Finally, the same *Co-1⁴ locus* was also significant for the CDP at the end of Pv01. The new SNPs identified,

**Funding:** This study was funded by the Fundação de Amparo à Pesquisa do Estado de São Paulo (FAPESP), under doctoral scholarship conceded to CPA author (Proc. 2019/19670-2) and research grants (Proc. 2017/24711-4) conceded to LLBR author.

**Competing interests:** The authors have declared that no competing interests exist.

especially those associated with more than one race, present great potential for use in marker-assisted and early selection of inbred lines.

## Introduction

Approximately 70 species have been described in the *Phaseolus* genus, only five of which are cultivated; *Phaseolus vulgaris* L. (bean) is considered the most important species in the *Phaseolus* genus for direct consumption in the human diet [1–3]. The species arose in Mexico [4,5], from where it spread to South America, giving rise to two distinct gene pools, called Mesoamerican and Andean [6,7]. In some African and American countries, beans are responsible for providing an average of 15% of total daily calories and 36% of protein content consumed [8]. World bean consumption has increased in recent years, with dry bean production reaching about 30 million tons in most recent analysis [9]. Most beans are produced by the countries of Asia and the Americas, which together account for approximately 75% of world production. Brazil stands out as one of the largest producer and consumer of beans in the world, and is responsible for 36% of the production on the American continent [9].

The estimated production of the 2020/2021 common bean crop in Brazil was about 3.2 million tons [10], with the carioca bean variety accounting for 70% of common bean consumption, followed by the black bean variety with 15% [11]. The first carioca cultivar was released in 1971 and, due to high yield (i.e., superiority of approximately 35% in relation to the varieties launched in the 60s), the new cultivar Carioquinha quickly came to predominate bean growing and consumption in Brazil [12]. The carioca commercial group cultivars are characterized by cream-colored grain with brown stripes and high yields [13] and they belong to the Mesoamerican gene pool [14]. Although the variety has high yield, it is far from its genetic potential since it is susceptible to diseases, such as anthracnose, which can cause losses of up to 100% of production [15].

Anthracnose caused by the ascomycetous *Colletotrichum lindemuthianum* (Sacc. and Magnus) Briosi and Cavara and considered to have a high pathogenicity, is characterized by small brown spots throughout the aerial part of the plant, frequent beginning in leaf veins, stems, and petioles [11,12]. Due to the high pathogenic variability of the fungus, Pastor-Corrales [16] proposed the classification of *C. lindemuthianum* isolates into physiological races based on the susceptible and resistant responses of the isolates to a differentiating series composed of 12 differential cultivars of common bean. A total of 1,590 isolates have already been characterized, resulting in identification of 182 races worldwide [15]. In Brazil, 474 isolates have been tested and 60 races described, with race no. 65 as the most frequent [15]. More specifically for the state of São Paulo, a study characterized 51 isolates and identified 10 races, with a predominance of race 65 and 81. The study furthermore indicated race 321 and 351 as the most pathogenic races [17]. Coelho et al. [18] also reported 65, 73, and 81 as some of the most frequent occurring races in Brazil.

The development of cultivars with durable resistance to disease is the most economical, efficient, and environmentally friendly resource as it avoids intensive use of pesticides, ensuring higher yields and less environmental contamination. [19]. For that reason, cultivars with resistance to multiple races and the respective genes associated with this resistance need to be investigated and identified, enabling their use in common bean breeding programs. About 25 *loci* with multiple alleles of resistance to ANT, from both Andean and Mesoamerican origin, have already been identified [20–22]. The *loci* with the greatest and dominant effect are

designated "*Co*". The *loci* of Mesoamerican origin include *Co-2*, *Co-3* (*Co-3²*, *Co-3³*, *Co-3⁴*, and *Co-3⁵* alleles) *Co-4* (*Co-4²* and *Co-4³* alleles), *Co-5* (*Co-5²* alleles), *Co-6*, *Co-11*, *Co-16*, *Co-17*, *Co-u*, and *Co-v*, mapped on chromosomes Pv02, Pv03, Pv04, Pv07, Pv08, and Pv11, respectively [23–35]. The "*Co*" *loci* originating from Andean origin are *Co-1* (*Co-1²*, *Co-1³*, *Co-1⁴*, and *Co-1⁵* alleles), *Co-12*, *Co-13*, *Co-14*, *Co-15*, *Co-x*, *Co-w*, *Co-y*, *Co-z*, *Co-Pa*, *Co-AC*, *and CoPv01^{CDRK}* on chromosomes Pv01, Pv03, and Pv04 [30,31,35–49].

Other studies involving QRL mapping have also been reported. Lopéz et al. [50] identified five QRL on three chromosomes; Zuiderveen et al. [51] reported 14 QRL with greater effects on chromosomes Pv01, Pv02, and Pv04, and 2 others with lesser effects on Pv10 and Pv11. Wu et al. [52] mapped 9 QRL (on Pv01, Pv02, Pv04, Pv05, Pv06, Pv10, and Pv11) and Perseguini et al. [53] another 17 QRL (on Pv01, Pv02, Pv03, Pv04, Pv05, Pv06, Pv07, Pv08, and Pv11). Recently, Fritsche-Neto et al. [54] identified a QRL with a greater effect on Pv02, explaining 25% of the resistance observed for nine environments evaluated.

In addition to the resistance *loci*, several studies have reported many ANT resistance sources from mainly cultivars of Andean origin [43,45,48,49,55–58], and also some of Meso-american origin, belonging to the black or special bean commercial classes [30,33,34,40,59–63]. Currently, developing new carioca cultivars through breeding is still a challenge as the variety demands high grain quality [64]. Breeding programs routinely exploit only the elite germplasm, which results in a narrow genetic base [65]. It is difficult to use Andean accessions as a source of resistance due to the reproductive incompatibility during the hybridization [66]. Recently, Almeida et al. [67] demonstrated that it is feasible to use Andean accessions to improve inbred lines resistance to angular leaf spot and that marker-assisted selection (MAS) is an important tool However, obtaining elite genotypes was only possible after backcrossing cycles. Thus, identification of *loci* associated with ANT resistance in carioca bean is extremely important in development/breeding of elite genotypes with resistance to multiple races.

The current study aimed at identifying carioca genomic regions associated with race-specific ANT resistance and recommendation of these cultivars in Brazil will lead to genetic improvement of the common bean. A CDP, previously validated for GWAS, was characterized in terms of resistance to three different physiological ANT races (65, 81, and 321) and genotyped using high-throughput genotyping technologies (i.e., BeadChip BARCBean6K_3). The results were corroborated by linkage mapping using a segregating carioca population derived from the cross between the IAC-Milênio and AND-277 cultivars.

## Materials and methods

### Plant material and high-throughput genotyping

The CDP used in the present study was composed of 125 bean genotypes, including landraces, advanced inbred lines, and commercial cultivars. The set represents genotypes from the main genetic breeding programs in the country, with cultivars released by 11 different institutions. These vary from the first cultivar carioca released by the Instituto Agronômico—IAC (Campinas, SP, Brazil) in 1971 (i.e., Carioquinha or Carioca comum) to more recent cultivars, such as IPR-Sábia and IAC-1850, released in 2017 and 2019, respectively. The panel was genotyped using the BeadChip Illumina technology by BARCBean6K_3, with 5,398 SNPs [68], and validated for GWAS by Almeida et al. [69] and successfully used to identify QRL associated with angular leaf spot by Almeida et al. [70]. All the information on the CDP is given in S1 Table of the Supplementary Materials, including the phenotypic and genotypic data.

The quality of the genotypic data was analyzed using the Genome Studio 2.0 software (Illumina), which filtered markers with call frequency and GenTrain score < 0.6. The TASSEL 5.0 software [71] was used to eliminate SNPs with minor allele frequency < 3%,

heterozygosity > 5%, and missing data > 10%. The high-quality genotypic matrix was converted into a HAPMAP file format, with the reference allele represented by "A", the alternative allele by "G", the heterozygous by "R", and missing data by "N". The BARCBean6K_3 was developed based on the first common bean genome (i.e., *Phaseolus vulgaris* v1), and therefore, the flanking sequences of each SNP were blasted (e.g., BLASTN) against the most current reference genome, *Phaseolus vulgaris* v2.1 [8], and the position of each SNP was obtained. Markers of unknown position in the genome were removed, and "N" *loci* were imputed using the Beagle 5.0 software [72].

For linkage mapping, the AM genetic map estimated from 1,114 SNPs genotyped by GBS (genotyping by sequencing) in the AM segregating population $BC_2F_3$ ({[(♀AND-277 × ♂IAC-Milênio) × IAC-Milênio] × IAC-Milênio}) was used. The AM population was developed and validated for genetic mapping by Almeida et al. [70] and was composed of 91 inter-pool lines selected according to the carioca grain ideotype. All the information on the genotypes of the AM population is presented in S1 Table in the Supplementary Materials, including the phenotypic and genotypic data.

## Phenotypic evaluation

For evaluation of resistance, monosporic cultures of physiological races 65 (IAC fungal library code14781), 81 (IAC fungal library code 14786), and 321 (IAC fungal library code 14831) of *C. lindemuthianum* were selected due to their importance for improvement of the variety as these are the most frequent occurring races in Brazil. The respective races were characterized by Ribeiro et al. [17] according to the differentiating series proposed by Pastor-Corrales [16]. Both sets, the CDP and the AM population, were evaluated individually for the severity of their reaction to each race, following a randomized block experimental design with three replications. The plot consisted of one row with five plants randomized in trays ($36 \times 26 \times 7$ cm) filled with sterile vermiculite.

A total of 15 seeds per genotype were pre-germinated in germination paper (Germitest ®) in BOD (Biochemical oxygen demand) at 25°C for three days. On the third day, the seeds were transplanted and grown in the greenhouse for nine days. The trays were then placed in a chamber with controlled humidity and temperature. The plants were inoculated by spraying on both leaf surfaces (concentration of $1.2 \times 10^{-6}$ spores/mL$^{-1}$). The inoculum was produced according to the method proposed by Cárdenas et al. [36] using test tubes containing sterile pods partially immersed in an agar-water medium.

The inoculated plants were maintained in high humidity (> 95%) for 48 h at 22°C, and disease severity was assessed 10 days after inoculation (DAI). The diagrammatic scoring scale proposed by Gonzáles et al. [73], based on the severity of the symptoms resulting from inoculation with ANT races, was used for evaluation. Plants with scores from 1.0 to 3.0 were considered resistant, from 3.1 to 6.0 were considered moderately resistant, and from 6.1 to 9.0 were considered susceptible. In all trials, the cultivar IAC-1850 was used as a resistance check, due to its high resistance to all races [74], and Rosinha G2 was used as a susceptible check [17].

## Statistical analysis

Analysis of variance was performed for all phenotypic evaluations performed through the ExpDes package [75], and genetic parameters including selective accuracy which measures the precision of the experiment, were estimated by the RBio software [76]. The genetic resistance correlation of CDP for three races evaluated was done by Pearson's correlation. For mapping analysis, genotypic values were estimated using the REML/BLUE (Restricted Maximum Likelihood/Best Linear Unbiased Estimator) model by the Be-Breeder package [77].

The Fixed and random model Circulating Probability Unification (FarmCPU) [78] implemented in the GAPIT 2.0 package [79] was used for association mapping due to its high statistical power and greater sensitivity to QRL with lesser effects. The FarmCPU uses the multi*locus* mixed model (MLM) and performs the analysis in two interactive steps: a fixed-effect model is applied first, followed by a random-effect model. Both models are repeated interactively until no significant SNP is detected. As demonstrated by Almeida et al. [14], the CDP does not require the use of a structuring matrix to correct type I errors (i.e., false positives), since there are no subgroups in the set. The *p* value threshold of each SNP for the first interactive steps in the model was determined by the permutation method using the function *FarmCPU.P.Threshold* (500 repetitions). The Bonferroni [80] threshold method (cutoff $\alpha =$ 0.05) was also used to determine significance in the Manhattan plot. In order to identify the QRL associated with the three races used together (i.e., the mixture of races), in addition to mapping the evaluations made, a fourth analysis was performed, considering the three evaluations together. For that reason, the genotypic value of each genotype was given by the highest adjusted average among the three ANT races used.

The AM genetic map, constructed by Almeida et al. [70], was used for the linkage mapping. The identification of QRL followed the approach described by the authors using the OneMap package [81]. Thus, the probabilities of the genotypes of the putative QRL were obtained by the hidden Markov chain, with steps every 1 cM. The adjusted values of the traits were used to fit the QRL mapping models in a progressive manner. First, markers with significant effects were identified using a fixed linear regression model, and the significant markers were used as cofactors in the composite interval mapping (CIM) model, proposed by Zeng [82]. The significance of the putative QRL were defined by the threshold obtained by 1,000 permutations [83], considering the significance level of 5%. The putative QRL identified were used as a starting point for MIM, based on the model proposed by Kao et al. [84]. The MIM mapping strategy involved three steps: searching for QRL, testing effects, and selecting the model. Starting from the model with the QRL identified in the CIM, new QRL were identified out, and the QRL with the highest likelihood of odds (LOD) values were inserted into the model. Then, all QRL were tested for conditional significance. Both models (complete and reduced) were also compared regarding the Akaike information criterion—AIC [85]. If a QRL had a non-significant effect or an AIC value higher than the reduced model, it was removed from the model. The procedure was repeated until no QRL was added or removed from the model. Thus, the final model was selected, and the positions and effects of the QRL were re-estimated, as well as the variation explained by each QRL ($R^2$).

## Candidate genes

The physical position of all the significant SNPs associated with resistance to ANT was used for a thorough search for candidate genes through genetic annotation using *Jbrowse* from Phytozome v11.0 [86] and the reference genome *Phaseolus vulgaris* v2.1 [8]. For the search, we considered a confidence interval window of 0.59 Mbp, the average distance identified by Almeida et al. [14] for the CDP (i.e., distance to LD decay = $r^2$ 0.2).

## Potential sources for bean resistance breeding

With the goal of selecting the genotypes that exhibit the best multiple race resistance and have favorable alleles, resistant accessions for all the races evaluated were selected, as well as the significant markers for GWAS. The genotypic data of each genotype with high resistance was converted into a GenAlEx file, through which a favorable allele (i.e., an allele that leads to an increase in resistance) of each SNP was represented by "RR", an unfavorable allele by "SS", and

a heterozygous marker by "RS". A control genotype containing all "RR" alleles was inserted in the genotype matrix. To select accessions based on significant SNPs, the set was discriminated and grouped by DAPC proposed by Jombart et al. [87] and implemented in the ADEGENET v2.1.1 package [88]. The DAPC analysis is considered free of Hardy-Weinberg and LD, and consists of transformation of genotypic data by the PCA into components that better explain the genetic variance, and these components are used for linear Discriminant Analysis [87]. The number of clusters required was two—one group for the resistant cultivars (i.e., grouped by the presence of favorable alleles) and another for the susceptible cultivars.

## Results

### Phenotypic resistance

Characteristic symptoms of the disease (i.e., necrotic lesions on leaf veins and petioles) were observed for all experiments, from 7–8 DAI. As expected, the check cultivars showed highly contrasting resistance for all trials, and the entire AM population showed resistance to race 321. In addition, 92% and 34% of the AM population had a score $\leq 3$ for race 65 and 81, respectively. In relation to the CDP, a total of 41.6% of the genotypes had high resistance to the three races (Fig 1A), with a higher degree of severity (40%) for race 81 (Fig 1B). The boxplot shows the difference in the degree of resistance of the CDP (Fig 1C), and it shows the smallest phenotypic variation for races 65 and 321.

Analysis of variance showed high phenotypic variability for all tests (p <0.001), except for the AM population evaluation with race 321. However, there was no significance for the block effect, indicating the possibility of using a completely randomized design for future trials. The selective accuracy of all trials ranged from 0.96 to 0.99, and broad sense heritability was from 0.93 for CDP in the evaluation with race 65 to 0.99 for the AM population with race 81. Considering the CDP, the Pearson correlation was significant ($p < 0.001$) and positive for the three races, with the highest correlation between race 65 and 81 (60%), followed by race 81 and 321 (46%) and race 65 and 321 (40%).

### Genotypic resistance

After SNP calling, 1,942 high-quality SNPs genotyped in the 125 accessions of the CDP by high-throughput genotyping technologies were used for association mapping. GWAS was performed for each of the three races individually, and a fourth association was performed using the adjusted average of the most pathogenic race for each genotype (mixture). Considering all analyses (Table 1), a total of 17 SNPs showed high significance (i.e., $p < 0.00002$) according to the Bonferroni test [80]. Among them, two were associated with race 65, six with race 81, five with race 321, and four with the mixture of races (Fig 2A). The QQ-plot represented the quality of the analysis through the good fit of the model used (Fig 2B).

On chromosome Pv01, two SNPs were significant for race 321: the first, SNP ss715648889, at the 43.062.012 bp position and the second, SNP ss715645285, at the 49.139.392 bp position. At 1.49 Mbp away from the SNP ss715645285 (Fig 2C), the SNP ss715645299 was associated with race 81, showing the presence of a QRL associated with the most pathogenic races in this region. On chromosome Pv02, the SNP ss715647887 (35,782,225 bp position) also showed significance for race 321, and a second SNP, ss715648710 (41,709,302 bp position), was associated with joint analysis (the mixture of races). Precisely at the beginning of the Pv04 chromosome, two SNPs had the greatest significance for the four analyses (Fig 2A), with the SNP ss715649427 (535,120 bp position) associated with race 65, and the SNP ss715642306 (373,157 position) associated with race 81, 321, and the mixture of races.

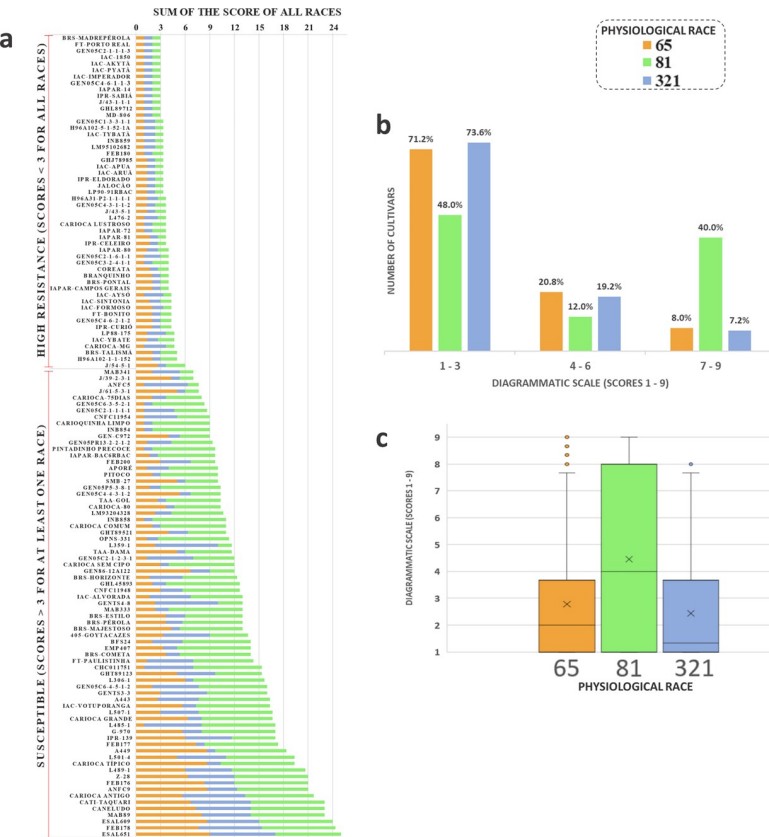

**Fig 1. Phenotypic results from evaluation of carioca diversity panel (CDP) resistance to multiple races of anthracnose.** (a) bar chart representing evaluation of the CDP regarding resistance to the *C. lindemuthianum* physiological races 65, 81, and 321. The bars represent the sum of scores of the phenotypic evaluation of each accession, with races 65, 81, and 321 being represented by colors (orange, green, and blue, respectively); (b) distribution of the number of accessions considered resistant (score ≤ 3), moderately susceptible (score from 4 to 6), and susceptible (score ≥7) for the three races evaluated; (c) boxplot for the three races evaluated.

The SNPs on the Pv04 chromosome were at 0.16 Mbp from each other (Fig 2C), making it possible to identify the presence of a unique and important QRL in this region. Considering the allelic substitution test, the alternative "A" allele of SNP ss715649427 was responsible for the increase in resistance of 0.8 for race 65 considering the 1–9 scoring scale, while the alternative "C" allele of SNP ss715642306 led to susceptibility of -1.8, -0.78, and -1.15 for races 81 and 321 and the mixture of races, respectively (Fig 2D). The results indicated an important QRL associated with multiple ANT races at the beginning of the Pv04 chromosome, at the 0.45 Mbp position.

A second SNP (ss715639578) on the Pv05 chromosome was significant for race 65 at 35,969,299 bp, and a single SNP (ss715646017, 37,347,362 bp position) on Pv07 was associated with race 81 and with the mixture of races. The Pv08 showed the highest number of associations; the first SNP (ss715647427, 15,431,263 bp position) was significant for race 81 and the second (ss715639361, 58,761,122 bp position) for the mixture of races. In the same region, at 3.2 Mbp, two other SNPs showed significance for race 81 (ss715646102, 61,452,776 bp position and ss715646115, 61,837,278 bp position) and one for the mixture of races (ss715639361, 58,761,122 position). Considering the last three SNPs on the same chromosome, the distance between them was not greater than 0.68 Mbp (Fig 3C), showing the presence of a single QRL associated with the races evaluated.

**Table 1. Genome-wide association study results for the association between phenotypic evaluation of the carioca diversity panel (CDP) with multiple races of anthracnose and the high-throughput genotyping.**

| Race | SNP | Chr | Position v2.1 | Ref. allele | Alt. allele | *p* value | MAF | Effect |
|------|-----|-----|---------------|-------------|-------------|-----------|-----|--------|
| **65** | ss715649427 | Pv04 | 535,120 | G | A | 1.8E-05 | 0.292 | 0.80 |
| | ss715639578 | Pv05 | 35,969,299 | T | C | 9.7E-06 | 0.096 | 1.14 |
| **81** | ss715645299 | Pv01 | 50,635,589 | C | T | 5.4E-06 | 0.076 | 1.74 |
| | ss715642306 | Pv04 | 373,157 | T | C | 7.2E-13 | 0.336 | -1.83 |
| | ss715646017 | Pv07 | 37,347,362 | C | T | 2.5E-10 | 0.476 | 1.47 |
| | ss715647427 | Pv08 | 15,431,263 | C | T | 5.5E-06 | 0.084 | -1.29 |
| | ss715646102 | Pv08 | 61,452,776 | G | A | 2.1E-05 | 0.232 | 0.84 |
| | ss715646115 | Pv08 | 61,837,278 | C | T | 2.9E-06 | 0.328 | -1.01 |
| **321** | ss715648889 | Pv01 | 43,062,012 | C | T | 2.5E-07 | 0.048 | 1.51 |
| | ss715645285 | Pv01 | 49,139,392 | C | T | 2.4E-06 | 0.344 | 0.60 |
| | ss715647887 | Pv02 | 35,782,225 | T | C | 1.3E-07 | 0.104 | 1.13 |
| | ss715642306 | Pv04 | 373,157 | T | C | 6.6E-08 | 0.336 | -0.78 |
| | ss715646756 | Pv08 | 62,139,343 | A | G | 9.8E-07 | 0.208 | 0.80 |
| **Mix** | ss715648710 | Pv02 | 41,709,302 | G | A | 5.9E-06 | 0.032 | -2.38 |
| | ss715642306 | Pv04 | 373,157 | T | C | 7.1E-07 | 0.336 | -1.15 |
| | ss715646017 | Pv07 | 37,347,362 | C | T | 8.0E-06 | 0.476 | 0.98 |
| | ss715639361 | Pv08 | 58,761,122 | T | C | 9.9E-06 | 0.192 | -0.90 |

Race: *C. lindemuthianum* physiological races; Chr: Chromosomes; Ref. allele: Reference allele; Alt. allele: Alternative allele; MAF: Minor allele frequency; Effect: Allelic substitution effect.

Significant for association mapping using the CDP evaluated phenotypically for resistance to *C. lindemuthianum* physiological races 65, 81, and 321.

Regarding QRL analysis using the AM population (Table 2), three QRL showed significance for race 65: the ALS3.1$^{AM}$ on chromosome Pv03 (estimated position 154 cM), explaining 9% of the phenotypic variance, flanked by "Marker244" and "Marker248"; the ANT10.1$^{AM}$ on

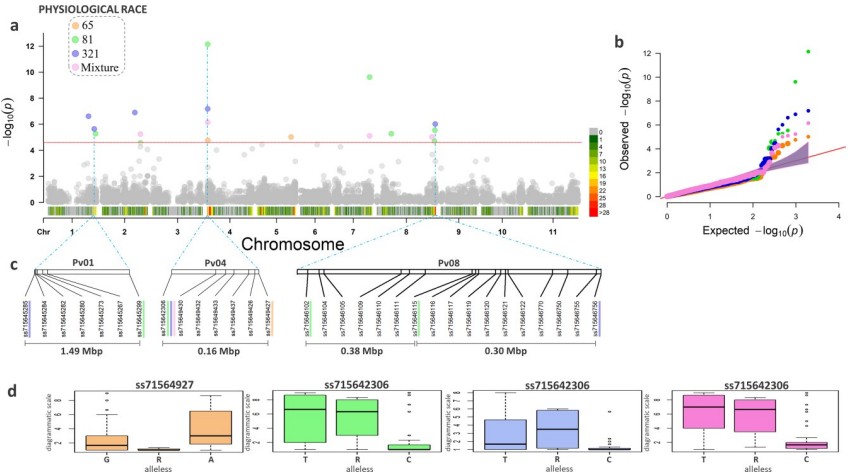

**Fig 2. Genome-wide association study analysis for resistance to multiple races of anthracnose with 1,942 high-quality simple nucleotide polymorphisms (SNPs) genotyped in the carioca diversity panel (CDP).** (a) Manhattan plots showing the association between the SNP markers and the physiological races of *C. lindemuthianum*. Each color corresponds to a different race, with the colors orange, green, blue, and pink corresponding to races 65, 81, 321, and admixture, respectively; (b) QQ-plot for the races evaluated; (c) Position of significant SNPs in genomic regions with SNPs associated with more than one race, showing the distance between them; (d) Boxplots illustrating the relationships between alleles and phenotypes for the significant SNPs located at the beginning of the Pv04 chromosome and associated with all the races tested.

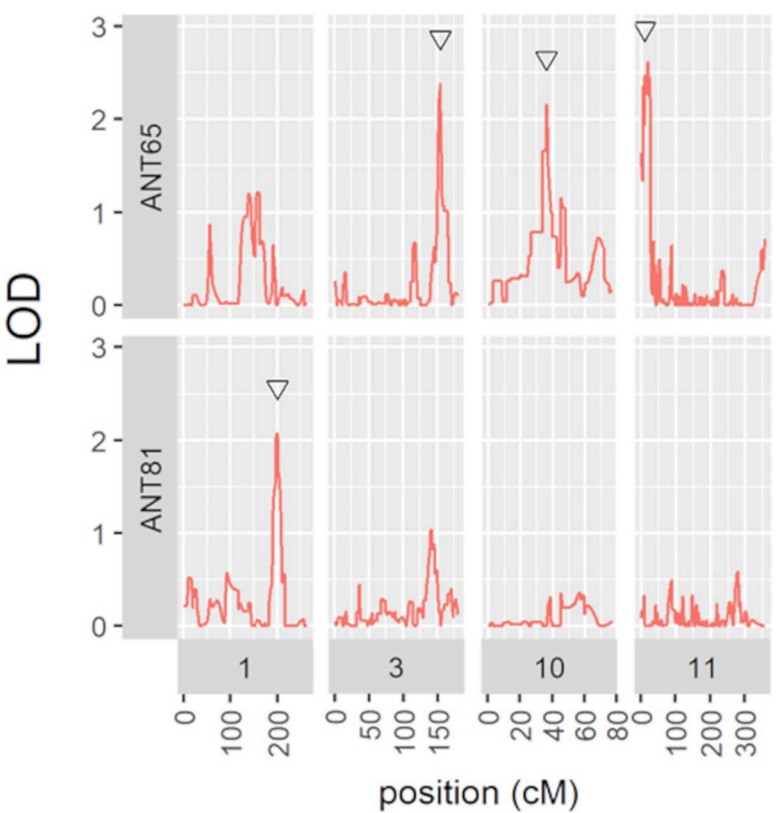

**Fig 3. Quantitative resistance *loci* (QRL) plots indicating the LOD score values for each marker position.**
Manhattan LOD scores obtained by multiple interval mapping analysis (y) using the molecular marker distances of the AM genetic map for *C. lindemuthianum* races 65 and 81. Black triangles indicate the position of significant QRL.

**Table 2. Quantitative resistance *loci* association results (QRL).**

| Race[1] | QRL[2] | Pv[3] | EPI (cM)[4] | Flanking Marker | Physical position[5] | LOD | R$^2$ (%) | AE[6] | SE[7] | *p* value |
|---|---|---|---|---|---|---|---|---|---|---|
| 65 | ANT3.1[AM] | 3 | 154.42 (151.31–156.29) | Marker244 | 18.14 | 2.37 | 9.12 | 0.59 | 0.18 | 0.001 |
| | | | | Marker248 | 22.93 | | | | | |
| 65 | ANT10.1[AM] | 10 | 35.89 (33.58–37.64) | Marker908 | 41.46 | 2.15 | 5.28 | 0.86 | 0.27 | 0.002 |
| | | | | Marker911 | 44.18 | | | | | |
| 65 | ANT11.1 [AM] | 11 | 8.98 (5.72–25.32) | Marker933 | 11.01 | 2.46 | 9.52 | 0.77 | 0.23 | 0.001 |
| | | | | Marker934 | 31.69 | | | | | |
| 81 | *Co-1*[4] | 1 | 199.91 (191.52–207.48) | Marker34 | 47.81 | 2.07 | 9.95 | 0.85 | 0.27 | 0.002 |
| | | | | TGA1.1 | 50.02 | | | | | |

[1]*C. lindemuthianum* race.

[2]QRL were named according to Pedrosa-Harand et al. [*81*].

[3]Chromosomes.

[4]Estimated position and interval QRL.

[5]Physical position (Mbp) according to the reference genome v2.1.

[6]Additive effect.

[7]Standard error.

Anthracnose QRL in multiple interval mapping for *C. lindemuthianum* races 65 and 81, using the segregating AM population (AND-277 × IAC-Milênio).

Pv10 (estimated position 35.89 cM), with $R^2$ estimated at 5% and flanked by "Marker908" and "Marker911"; and finally, the ANT 11.1[AM] on Pv11 (estimated position 8.98 cM) explaining 9.5% of phenotypic resistance. For race 81, a single QRL ANT 1.1[AM] at the end of chromosome Pv01 showed high significance, explaining 10% of the resistance and flanked by "Marker34" and "TGA1.1".

### Resistance genes

Considering the average LD decay of the CDP, the SNPs ss715646102, ss715646115, and ss715646756 on the Pv01 chromosome are in LD and belong to the same haplotype block, as well as the SNPs ss715642306 and ss715649427 on Pv04. Except for the SNPs ss715647887, ss715648710, ss715639578, ss715646017, ss715647427, ss715646115, and ss715646756, all the SNPs were close to the resistance genes (i.e., distance < 1 Mbp), whether R genes or genes encoding protein kinases (Fig 4).

The SNP ss715642306 (Pv04) significant for races 81 and 321 and for the mixture of races, which was in LD with the SNP ss715649427 (Pv04) significant for race 65, was in the first exon of the Phvul.004G005800 gene (functional notation: alkane hydroxylase cyp96a15). A total of 16 resistance candidate genes were identified for the range of both SNPs (i.e., with 14 R genes and two encoding protein kinases). Another cluster of genes associated with resistance was placed in the confidence interval of the SNP ss715645285 (Pv01) associated with race 321, with a total of six genes encoding protein kinases.

The second-largest cluster with protein-encoding kinase genes (i.e., six genes in a range less than 0.58 Mbp) was at 2.3 Mbp of the SNP ss715639578 (Pv05) significant for race 65. Another three genes encoding protein kinases are close to the significant SNPs at the end of the Pv08 chromosome (i.e., ss715639361, ss715646102, ss715646115 and ss715646756). The SNPs ss715647887 and ss715648710 on Pv02 were at the greatest distance from the candidate genes and were in the last intron of the Phvul.002G193800 gene (functional annotation: encoding urease accessory protein) and in the second intron of the Phvul.002G245000 gene (functional annotation: encoding ATP-dependent protease cereblon), respectively.

### Best cultivars: Phenotypic and genotypic resistance

Altogether, 52 cultivars showed high resistance to the three races in the phenotypic evaluations (Fig 1). To select the most promising accessions as sources of resistance for common bean breeding, DAPC was used and grouped 18 of the 52 accessions with 100% membership probability for the cluster containing the resistance pattern (i.e., genotype containing all RR alleles) when considering the 14 SNPs associated with ANT resistance (Fig 5A). The analysis did not show overlapping for both clusters (Fig 5B), except for the genotype H96A31, which presented only 20% of the membership probability for the green cluster (favorable alleles).

Among the 18 accessions clustered according to the presence of favorable alleles, the cultivars IAC-Akytã, IAC-Pyatâ, IAC-Imperador, IAC-Apuã, IAC-Tybatã, IAC-Aysó, IAC-Formoso, IPR-Curió, Carioca MG, and IAC-Ybaté which were most promising as sources of genetic resistance to ANT. Besides being resistant to the three ANT races, all of them are commercial cultivars with high yield potential [90].

## Discussion

Several studies aiming at the identification of markers associated with ANT resistance *loci* in common bean have already been conducted, leading to the identification of genotypes showingthe *loci* of greatest effect. However, most studies involve Andean accessions [30,33,34,40,59–63], and most of the studies with Mesoamerican accessions did not involve

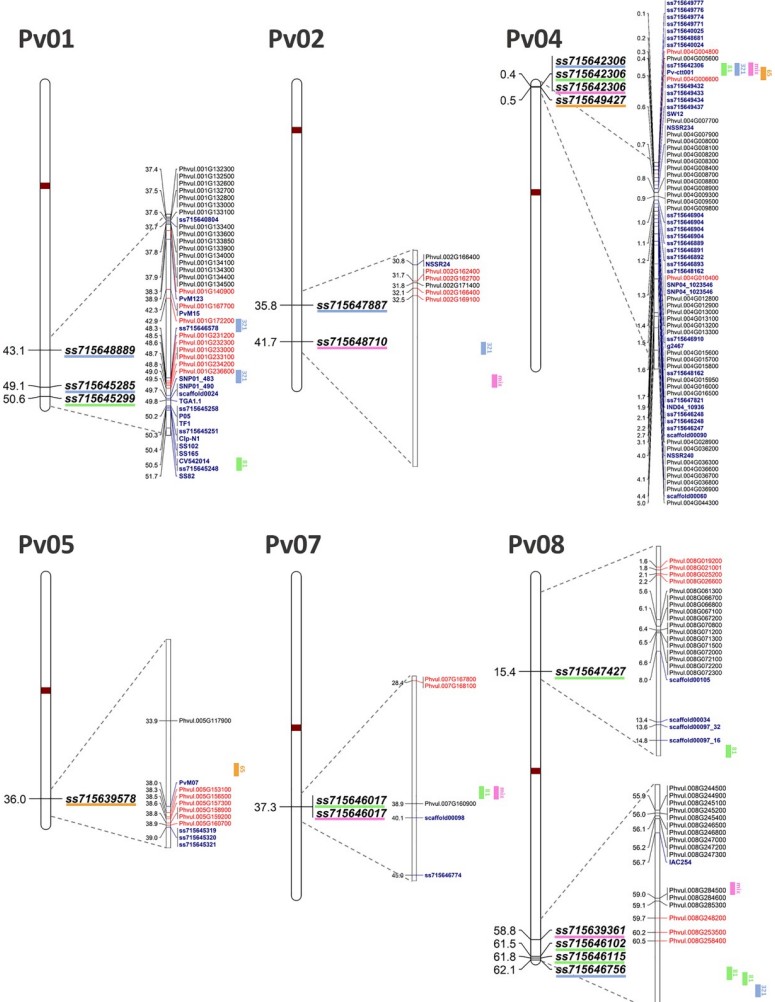

**Fig 4. MapChart showing the simple nucleotide polymorphism associated with multi-race anthracnose resistance loci.** Each chromosome represents the physical position of the simple nucleotide polymorphisms (SNPs) significant for the three races of anthracnose used independently and for the mixture of races in the genome-wide association study (GWAS). The significant quantitative resistance *loci* (QRL) are represented by orange, green, blue, and pink bars on the right side of the chromosomes for races 65, 81, 321, and the mixture of races, respectively. The part of the amplified chromosomes shows the physical position of the main genes involved in the disease resistance response: the R genes (containing nucleotide-binding and leucine-rich repeat domains—NB-LRR) in black and in a position according to Schmutz et al. [8], and the genes encoding kinases in red, according to Bisneta and Gonçalves-Vidigal [89]. Markers associated with the resistance *loci* found in previous studies are highlighted in dark blue. Centromeric regions are represented in brown and physical positions provided in Megabase pairs (Mbp), according to the reference genome: *Phaseolus vulgaris* v2.1.

carioca bean varieties [43,45,48,49,55–58]. Although studies on characterization of ANT resistance and selection of carioca inbred lines and cultivars have been conducted [91–93], no previous association mapping study was conducted with only carioca accessions using a linkage mapping approach with the carioca variety as the resistant parent.

The characterization of new sources of ANT resistance is extremely important for carioca been breeding. However, the use of Andean accessions to improve Mesoamerican cultivars is extremely difficult [94–96], mainly due to reproductive incompatibility [66] and the breakdown of epistatic interactions specific to each gene pool [97]. An inter-pool cross involving parents from different commercial classes is considered a bottleneck for breeders, mainly due

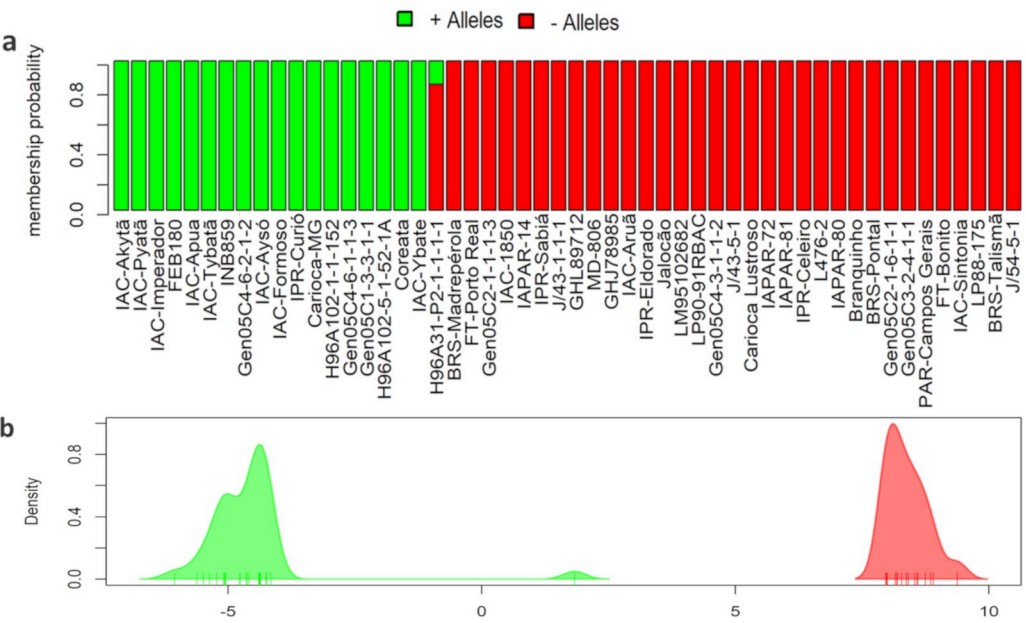

**Fig 5. Discriminant analysis of the principal components (DAPC) carried out using the simple nucleotide polymorphisms (SNPs) associated with resistance to anthracnose classified by the resistance allele.** (a) Compoplot showing clustering performed only with the 52 accessions considered resistant to all races (i.e., score ≤ 3), both groups being formed based on the frequency of resistance and susceptibility alleles, in green and red, respectively. (b) Graphical representation of the separation of both groups, showing the absence of overlapping for discriminant analysis.

to the segregation observed for the type of grain [98]. In the case of the carioca commercial class, the complexity is even greater since any change in the ideotype leads to a devaluation of the grain on the market. For that reason, Brazilian common bean breeding programs usually prefer crosses among elite carioca cultivars, due to the additive effects of quantitative traits, obtaining superior advanced bean lines [64,99].

Recent studies have shown that as a result of the limitation of the operational capacity of breeding programs, the right choice of parents for genetic improvement is extremely important [64,69,100]. Some authors have reported the possibility of selecting superior parents based only on genotypic information. However, for parental selection towards anthracnose resistance the best approach is to gather both phenotypic and genotypic information [64]. Therefore, the phenotypic characterization of the CDP for resistance to the main anthracnose races in the state of São Paulo (Brazil) and the genomic information on favorable alleles regarding ANT resistance *loci* represent a great advance in breeding of carioca common bean.

## Association mapping

In the present study, we identified many QRL associated with *C. lindemuthianum* race-specific resistance. Among them, the SNPs ss715648889 (Pv01) and ss715647887 (Pv02) were significant only for the most pathogenic race (race 321), whereas ss715639578 (Pv05) and ss715647427 (Pv08) for the less pathogenic races 65 and 81, respectively. SNP ss715646017 (Pv07) also showed significance only for race 81; however, considering the mixture of races, it was also significant. Recently, Mungalu et al. [21] confirmed the race-specific resistance pattern, identifying a total of 14 QRL with nine different ANT races. Yet only two QRL were significant for all the races tested.

The identification of *loci*/markers associated with multiple races of any pathogen is extremely relevant within the aim of increasing ANT resistance in bean breeding programs.

Such *loci*/markers can be used in marker-assisted selection (MAS) in an efficient way. From this perspective, the SNPs of greatest importance found in our study were ss715649427 and ss715642306 on the Pv04 chromosome, at 0.16 Mbp from each other and associated with the three of the races evaluated. We found 14 R genes and 2 encoding genes for protein kinases in the same region and within an interval of less than 0.6 Mbp (Fig 4). In addition to the candidate genes, a total of 14 markers have been associated with resistance *loci* in the same range as other studies, both by linkage mapping and GWAS [20,51,52,61,101]. In the same region, the *Co-3 locus* was identified by several authors, as well as its allelic series *Co-3²*, *Co-3³*, *Co-3⁴*, and *Co-3* [28,37,40,102,103]. However, among all the markers reported as associated with ANT resistance at the beginning of the Pv04 chromosome, the SNP ss715642306 showed greatest potential for use as a tool in MAS.

The same SNP ss715642306 that we identified as associated with the races evaluated was also mapped by Zuiderveen et al. [51] for race 7 in an Andean accession panel; by Costa et al. [101] for one of the five isolates characterized as race 65 on the Mesoamerican diversity panel; and by Valentini et al. [62] in a fine-mapping approach linked to the *Ur-14*, *Co-3⁴*, and *Phg-3 locus*, responsible for conferring resistance to rust, ANT (race 73), and angular leaf spot, respectively. The annotation procedure demonstrated that this SNP is within the exon of the Phvul.004G005800 gene that encodes the Cytochrome P450 protein, which is known to play an important role in the catalysis of redox reactions [104]. For its part, P450 can trigger a plant hypersensitive disease reaction by oxidative degradation [51,101].

Considering the QRL associated with more than one race (races 81 and 321), a second *locus* on Pv08 composed of three different significant SNPs (ss715646102, ss715646115, and ss715646756) showed an interval of 0.69 Mbp. In the same region (i.e., between the 59.1 Mbp and 60.5 Mpb positions), there are six genes involved in disease resistance mechanisms, with three genes encoding NB-LRR (nucleotide-binding, leucine-rich repeat, R genes) and three encoding kinase proteins (Fig 4). The NBS-LRR sequences represent a crucial resource for combating pathogen attack [105]. Perseguini et al. [53] also reported a marker at the 56.7 Mbp position significant for race 7. Oblessuc et al. [56] identified a marker (PF₅₃₃₀, 61.2 Mbp position) in the same region linked to the ANT08.3^UC QRL conferring resistance to race 38. The resistance *locus* linked to angular leaf spot (*Phg-2*), one of the five *loci* accepted by the Bean Improvement Cooperative Genetics Committee and the second identified in the Mesoamerican genotype [106], was also observed in the same genomic region [107].

Another QRL associated with more than one race (races 81 and 321) was identified at the end of chromosome Pv01 by the SNPs ss715645299 and ss715645285. In the same region on Pv01, between the 48.3 Mbp and 50.5 Mpb positions, is a large cluster of protein kinases -encoding genes that are involved in plant resistance mechanisms [89]. In the same interval, 15 markers were associated with ANT resistance in nine different studies [41,44,47,48,51,53,59,63,108]. In this region, several *loci* of greater effect have been reported, such as *Co-1*, *Co-x*, *Co-w*, *Co-14*, *Co-Pa*, *Co-AC*, and *CoPv01*^CDRK [89]. However, previous studies identified this resistance *loci* in Andean genotypes, while our study was the first to find it using an exclusively Mesoamerican diversity panel. The results reinforce the hypothesis that the different *loci* reported are a single large *locus* composed of a cluster of resistance genes and, depending on the race or genotypes used, different genes with large effects from the same cluster are activated, enabling the identification of different *loci* in the same genomic region.

## Linkage mapping

Unlike the PDC, the AM population is a set with resistance *loci* of Andean origin, since the AND-277 cultivar resistant to ANT was used as a donor parent to obtain the population. In a

previous study, the cultivar AND-277 showed resistance to races 64, 65, 73, 81, 87, 89, 119, 453, and 2047. In addition, it carries an allele of the *Co-1 locus* designated *Co-1$^4$* [109]. Furthermore, our results confirmed that AND-277 is resistant to races 65 and 81 and that the cultivar is also resistant to race 321. Gonçalves-Vidigal et al. [63] mapped a single and dominant *locus* on Pv01 in cultivar AND-277 flanked by the CV542014450 and TGA 1.1$_{570}$ markers, both associated with races 73 and 2047. The same *Co-1 locus* was identified in the present study for race 81 also flanked by the marker TGA1.$_{1570}$. However, the *locus* was not significant for race 65, demonstrating that the *locus* is probably related to more pathogenic ANT races (e.g., 73, 81, and 2047).

For race 65, three QRLs were considered significant, one on each of the Pv03, Pv10, and Pv11 chromosomes. The QRL ANT3.1$^{AM}$ on Pv03 was in the reference genome v2.1 between the 18.14 Mbp and 22.93 Mpb physical positions. On the same chromosome, two *loci* of greater effects have been reported in previous studies. The first *Co-13 locus* was identified by Lacanallo et al. [58] in the Andean landrace Jalo de Listras Pretas, linked to the marker OV20$_{680}$, followed by the second *Co-17 locus*, identified by Trabanco et al. [33] in the Mesoamerican cultivar SEL1308, linked to the marker NDSU_IND_3_0.0441. In the same region, Peseguini et al. [53] identified the IAC-167 marker at the 13.43 Mbp position, associated with both ANT and angular leaf spot disease. Vaz Bisneta et al. [110] also identified SNP S03_13038972 at the 13.38 Mbp position, associated with the ANT 1545 race.

The ANT10.1$^{AM}$ QRL was mapped between the 33.58–37.64 cM positions on the linkage map, and 41.46–44.18 Mpb on the physical map (reference genome v2.1). Mungalu et al. [21] mapped the ANT10.1$^{AS}$ QRL significant for ANT race 1331 in a nearby position, between the 39.2–40.6 cM positions on the AS map (Solwezi × AO-1012-29-3-3). The SNP ss715648593 was the closest marker to ANT10.1$^{AS}$ and was at 1.14 Mbp from Marker908 linked to ANT10.1$^{AM}$. The presence of both QRL at the same position corroborates the presence of a single Andean *locus* of lesser effect at the end of the Pv10 chromosome. On Pv11, we identified the QRL ANT11.1$^{AM}$ at the 8.98 cM position, a region in which the *Co-2 locus* in the Cornell cultivar was also mapped, by Geffroy et al. [29]. Rodríguez-Suárez et al. [111] identified another *Co-2$^{A252}$ locus* in the A-252 cultivar and, more recently, Campa et al. [55] identified the *Co-2$^{AB136}$ locus* on the AB-136 genotype. Several other studies involving GWAS have also identified markers associated with ANT resistance on Pv11 [51–53,89].

## Common bean breeding for anthracnose resistance

Common bean has two main centers of origin [5], and in the case of some diseases such as angular leaf spot, due to the process of co-evolution, the pathogens have the same classification and may show a higher degree of pathogenicity to a specific gene pool [112,113]. In the case of ANT, it is common to identify some QRL specific to a certain gene pool. *Co-1* is an example of a *locus* mapped exclusively in Andean cultivars, having been identified in Michigan Dark Red Kidney (*Co-1*), Kaboon (*Co-1$^2$*), Perry Marrow (*Co-1$^3$*), AND-277 (*Co-1$^4$*), Widusa (*Co-1$^5$*), Xana (*Co-1$^x$*), Pitanga (*Co-14*), Jalo EEP 558 (*Co-x*), Hongyundou (*Co-1$^{HY}$*), Amendoim Cavalo (*Co-AC*), California Dark Red Kidney (*CoPv01$^{CDRK}$*), and Paloma (*Co-Pa*) [89]. Our results using the AM population validated the presence of *Co-1$^4$* in the Andean AND-277 cultivar; however, a *locus* in the same position showed a significant association with the resistance of the carioca panel.

In both cases, the *loci* were not significant for race 65, clearly showing that it is a single QRL for both sets. The identification of the resistance *locus* of Andean origin in the Mesoamerican group corroborates the results recently presented by Almeida et al. [14]. The authors identified a number of SNPs with 100% contrasting alleles between the Andean and Mesoamerican

genotypes, which enabled accurate identification of Andean allelic introgression events in Mesoamerican cultivars. They showed that the SNPs with the most prevalent Andean allele in the Mesoamerican group were close to the main *loci* associated with disease resistance, confirming the use of Andean genotypes as a source of resistance for the genetic improvement of new Mesoamerican cultivars. In the study, among the four SNPs found on Pv01 with the highest frequency of the Andean allele in the Mesoamerican cultivars, three were exactly at the position of the *locus* that we have identified in both the AM population and the CDP (i.e., position at 48.5 Mbp).

Our results show that although Andean accessions are extremely important for genetic improvement aiming to gain ANT resistance, it is possible to select elite cultivars as sources of resistance for obtaining new inbred carioca lines, without the need to face the challenge of using parents from another gene pool and/or commercial class. In this sense, the commercial cultivars IPR-Curió, Carioca-MG, IAC-Akytã, IAC-Apuã, IAC-Aysó, IAC-Imperador, IAC-Pyatã, IAC-Formoso, IAC-Ybaté and IAC-Tybatã showed the greatest potential for use by Brazilian common bean breeding programs. These cultivars not only fulfilled the minimum requirements of any commercial cultivar [114–116], but also showed high resistance (i.e., score < 3) to the three races evaluated. Molecular evaluation by DAPC, estimated with the significant SNPs from GWAS, showed that these cultivars have the highest number of favorable alleles among the 18 genotypes grouped in terms of resistance pattern (Fig 5). Among them, the cultivar IAC- Pyatã also showed high resistance to angular leaf spot in a recent study [69] and had the highest average yield among the 11 other commercial cultivars tested by Pompeu [116].

In addition, with recommendation of the cultivars, the markers associated with the resistance *loci* have considerable potential for screening germplasm and MAS. Recently, Paulino et al. [117], using the resistant cultivar IAC-Formoso, showed the efficiency of marker-assisted backcrossing. After two cycles of backcrosses with selection carried out by nine markers associated with QRL associated with ANT resistance by Oblessuc et al. [56], the authors obtained isogenic advanced lines with superiority to the recurrent parent BRS-Pérola in terms of ANT resistance and fusarium wilt and also with greater yield and tolerance to seed coat darkening.

## Supporting information

**S1 Table. Details of the 125 carioca cultivars: Name, grain size (mm), commercial classification, institution of origin, adjusted mean (BLUE) of the resistance evaluation to the anthracnose (physiological race of ANT 65, 81, and 321), genealogy, and genotypic matrix.** (XLSX)

## Acknowledgments

We are grateful to Dr. Qijian Song of the Soybean Genomics and Improvement Laboratory, US Department of Agriculture–Agricultural Research Service (USDA-ARS) for support in genotyping. We are also grateful to Wesley Alexandre da Silva and Carlos Aparecido Fernandes for supporting the experiments.

## Author Contributions

**Conceptualization:** Caléo Panhoca de Almeida.

**Data curation:** Caléo Panhoca de Almeida, Jean Fausto de Carvalho Paulino, Gabriel de Moraes Cunha Gonçalves, Roberto Fritsche-Neto, Sérgio Augusto Morais Carbonell, Alisson Fernando Chiorato, Luciana Lasry Benchimol-Reis.

**Formal analysis:** Caléo Panhoca de Almeida, Jean Fausto de Carvalho Paulino, Caio Cesar Ferrari Barbosa, Gabriel de Moraes Cunha Gonçalves, Roberto Fritsche-Neto, Luciana Lasry Benchimol-Reis.

**Funding acquisition:** Luciana Lasry Benchimol-Reis.

**Investigation:** Gabriel de Moraes Cunha Gonçalves.

**Methodology:** Caléo Panhoca de Almeida, Jean Fausto de Carvalho Paulino, Caio Cesar Ferrari Barbosa, Gabriel de Moraes Cunha Gonçalves, Luciana Lasry Benchimol-Reis.

**Project administration:** Luciana Lasry Benchimol-Reis.

**Supervision:** Sérgio Augusto Morais Carbonell, Alisson Fernando Chiorato, Luciana Lasry Benchimol-Reis.

**Validation:** Jean Fausto de Carvalho Paulino, Sérgio Augusto Morais Carbonell, Alisson Fernando Chiorato, Luciana Lasry Benchimol-Reis.

**Visualization:** Jean Fausto de Carvalho Paulino, Roberto Fritsche-Neto, Sérgio Augusto Morais Carbonell, Alisson Fernando Chiorato, Luciana Lasry Benchimol-Reis.

**Writing – original draft:** Caléo Panhoca de Almeida.

**Writing – review & editing:** Jean Fausto de Carvalho Paulino, Caio Cesar Ferrari Barbosa, Gabriel de Moraes Cunha Gonçalves, Roberto Fritsche-Neto, Sérgio Augusto Morais Carbonell, Alisson Fernando Chiorato, Luciana Lasry Benchimol-Reis.

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
