## [Decision Letter · Decision Letter 0]

29 Mar 2021

PONE-D-21-03824

Genome-wide association mapping reveals race specific SNP markers associated with anthracnose resistance in carioca common beans

PLOS ONE

Dear Dr. Almeida,

Thank you for submitting your manuscript to PLOS ONE. After careful consideration, we feel that it has merit but does not fully meet PLOS ONE’s publication criteria as it currently stands. Therefore, we invite you to submit a revised version of the manuscript that addresses the points raised during the review process.

ACADEMIC EDITOR:

Dear Authors, Based on the reviewers' reports I have received, I decided that your manuscript may be acceptable for publication in PLOS ONE pending a major revision. Please see the comments form the reviewers below and make sure that all of them are addressed in your revision. I look forward to receiving it. Thank you.

We look forward to receiving your revised manuscript.

Kind regards,

Istvan Rajcan, Ph.D.

Academic Editor

PLOS ONE

Journal Requirements:

1. Please ensure that your manuscript meets PLOS ONE's style requirements, including those for file naming. The PLOS ONE style templates can be found athttps://journals.plos.org/plosone/s/file?id=wjVg/PLOSOne_formatting_sample_main_body.pdf and https://journals.plos.org/plosone/s/file?id=ba62/PLOSOne_formatting_sample_title_authors_affiliations.pdf

Additional Editor Comments (if provided):

Reviewers' comments:

Reviewer's Responses to Questions

**Comments to the Author**

1. Is the manuscript technically sound, and do the data support the conclusions?

Reviewer #1: Yes

Reviewer #2: Yes

2. Has the statistical analysis been performed appropriately and rigorously? 

Reviewer #1: Yes

Reviewer #2: Yes

3. Have the authors made all data underlying the findings in their manuscript fully available?

Reviewer #1: Yes

Reviewer #2: Yes

4. Is the manuscript presented in an intelligible fashion and written in standard English?

Reviewer #1: Yes

Reviewer #2: Yes

5. Review Comments to the Author

Reviewer #1: In general, the manuscript is poorly written with many grammatical errors and repetition of some phrases throughout the manuscript. The authors are strongly encouraged to re-write/thoroughly edit the manuscript. I have listed some specific examples below:

Abbreviations Define abbreviations upon first appearance in the text.

L20 In the present study, using a carioca diversity panel (CDP) and a carioca segregating population (AM population), both genotyped using high-throughput genotyping technologies, we employ multiple interval mapping (MIM) and genome-wide association studies (GWAS) for resistance to the main ANT physiological races present in the country. Did you use MIM-Gene or any other software? The sentence is too long.

L21 “study, using a carioca diversity panel (CDP, N=?) (state the number of lines in this panel) and a carioca segregating population (AM population)” Define abbreviations upon first appearance in the text (AM).

L22 “genotyped using high-throughput genotyping technologies,” Please name the genotyping technology used in your study.

L22 we employ multiple interval mapping (MIM) and genome-wide association studies (GWAS) for resistance to the main ANT physiological races present in the country. “Multiple interval mapping (MIM) and genome-wide association studies (GWAS) were used as mapping tools for the resistance genes to the major ANT…….

L24 In general, 14 SNPs “Define the abbreviations upon first appearance” and loci (Name the loci) associated with multiple races were also identified.

L27 “known to be related to plant immunity were identified in the confidence interval (please clarify which confidence interval (CI) you are referring to, i.e., is it on Pv 04. Rephrase the sentence.

L27 Through discriminant analysis of principal components (DAPC), cultivar sources with high potential for durable anthracnose resistance were recommended. Do you mean that you used DAPC to identify cultivars with high anthracnose resistance?

L29 The MIM confirmed the presence of the Co-1 4 29 locus in the AND-277 30 cultivar “remove “but showed that the locus” which revealed that it was the only one associated with resistance to ANT race 65.

L31 This is the first study to “delete point out” and insert “identify” new resistance loci in the AND-277 cultivar.

L33 The new SNPs identified, especially those associated with more than one race, “remove -presented” present great potential for use in {Remove-the process of” marker-assisted (remove- selection) and early selection of inbred lines.

Introduction

L47 “the largest producer (remove “s “) and consumer (remove “s “) of beans in the world,

L49 The estimated production of “the last “list the year when this data was collected/reported) common bean crop in Brazil

L52 “of approximately 35% in relation to the varieties of the period)” which period? Be specific.

“the new variety quickly came to” Name the variety and rephrase the sentence.

L53 “The (remove this-cultivars of the) carioca commercial group “add –cultivars” are characterized by cream-colored grain with brown stripes and high yield “add yields) (9) and ( and “they “belong to the

L67 “races 65, 73, and 81 as those of most frequent occurrence in 68 Brazil “replace with “as some of the most frequent occurring races in Brazil.

L69 The development of cultivars with “replace lasting, with durable” resistance ………..

L70 “ensuring the “replace grower greater economic yield” with “higher yields” and less environmental contamination. (16).”

L73 Delete “Up to this time, (~ ) “ What does this symbol stands for, the symbol should be described the first time it is used. “Insert “About “ 25 loci with multiple alleles of…….” both of” replace with “from both” Andean and Mesoamerican origin

L75 “Remove “These” and replace with “The” loci of Mesoamerican origin “Delete “are” and insert “include” Co

L78 Pv11 “Add, respectively” (20,21,30–32,22–29). The “Co” loci “delete “of “and replace with “originating from “Andean origin.

L88 reported many “remove “cultivar sources of “Add ANT resistance sources from mainly cultivars of Andean origin (40,42,45,46,52–55) “delete;”, and also some of Mesoamerican origin,

L90 “delete” “they belonged” add “belonging to the black or special bean commercial classes.

L91 Replace “Obtain” with developing

L92 Delete” For that reason”, “Delete tend to” i.e., Breeding programs routinely exploit only the elite germplasm “delete “of the variety”, which results in a narrow genetic base.

L93 Delete “The difficulty is even greater when it is necessary to Investigate” and replace with “It is difficult to use Andean accessions as a source of resistance (why?). Delete “For that reason”, “delete the” identification of loci associated with ANT resistance in carioca bean is extremely important “delete “with the goal of achieving” add” in development/breeding of elite genotypes with resistance to multiple races.

L98 resistance and (delete so, add) recommendation of these cultivars, “delete, that are resistant to the main ANT races” in Brazil (delete- “alleles favorable” and “add”) will lead to genetic improvement of the common bean (delete crop).

L99 Delete “For that purpose.”

L101 “Using high-throughput genotyping technologies”, name which genotyping technologies?

L103 Why use the two parents? Was there any unique trait etc.?

Materials and Methods

L108 Delete “the efforts of” add “genotypes from the main genetic breeding.

L109 The sentence is too long (L108-L111) Put a full-stop after 11 different institutions. These vary from the first…..

L127 The Am population was not is

L140 Plot (delete was composed) consisted of one row with five plants

L142 Delete of each, add per

L144 rephrase remove “remaining”, The seeds were transplanted and grown in the green house for nine days. Indicate the green house conditions i.e., humidity and temperature.

L150 replace intensity with severity, replace “generated by the infection of the disease” with resulting from inoculation with ANT races”

L153 Delete cultivar (used as a resistant check)

L154 Delete cultivar

L156 Rephrase, Analysis of variance was used on all phenotypic data.

L157 Delete, all and add, three races evaluated was done.

L178 which software was used for CIM?

L182 Replace “Starting from with Using the QRL model identified in CIM, replace “sought” with identified

L184 What does conditional significance mean? Both models (complete and reduced) what does this mean/difference

L207 Replace “best cultivars” with resistant cultivars vs susceptible cultivars (not worst)

L221 How did you measure the accuracy of the trials

L294 The LOD score for each QRL is below the threshold of LOD= 3.0 and they seem to explain less than 10% of phenotypic variation indicating they are minor ones. How many environments and years was this replicated?

L295 Name the C. lindemuthianum races used. The table should be stand alone/self explanatory

L312 A total of 16 resistance candidate genes were identified for the range of both SNPs, with two encoding protein kinases and the remaining R genes. This is not clear, clarify.

L346 which were most promising, delete “as sources of genetic resistance to ANT”. In addition to”,

L347 Beside being resistant to the three ANT races, “delete” “and having been grouped in the resistance cluster (i.e., favorable alleles)”, all of them are commercial …..

L359 identification of genotypes “holding” rephrase

L372 “programs tend to prefer to perform” replace with tend to do crosses among elite carioca cultivars

L378 may be sustained by both data sets, which ones?

Reviewer #2: Reviewer comments for “Genome-wide association mapping reveals race specific SNP markers associated with anthracnose resistance in carioca common beans.”

General Comments

The introduction is very informative but leaves room for questions that are not addressed and/or clarified until the discussion. It may be beneficial to sufficiently address the topic of Andean vs. Mesoamerican gene pools in the introduction and then relate it to this particular study within the discussion section or hold off to address it in detail in the discussion session. Currently it is difficult to follow divided between the two sections.

Minor comments and suggestions

- First sentence and second sentence, line 16 - 18

o Flip the order of the first and second sentence for clarity and re-organize wording of first sentence.

o Suggestion:

“Brazil is the largest consumer of dry edible beans in the world, 70% of consumption is of the carioca variety. (Introduce the impacts of anthracnose on these bean crops, what does it do?) The most effective strategy to overcome anthracnose (ANT), a disease caused by the fungus Colletotrichum Lindemuthianum, is the development of resistant cultivars.”

- Line 20

o Omit “In the present study”

- Line 46

o Omit “In this scenario”

- Line 49

o It is unclear what sentence means, “estimated production of the last common bean crop in Brazil…”. Was this the harvest of a particular year?

- Line 51

o It is a bit unclear what is meant by the use of the word “superiority” in the context of this sentence.

- Line 55-56

o The abstract would benefit from adding the sentence, “Although the variety has high yield…”

- Line 58 and 60

o Redundant to include “high degree of pathogenicity” twice in back to back sentences

- Line 61

o In scientific literature referring to anthracnose race identification, common wording I have seen is “susceptible” and “resistant” as opposed to “compatible” and “incompatible”. It may be good to stick to the re-occurring terminology.

o (Perhaps also beneficial to refer to Pastor- Corrales here in this explanation of race identification)

- Line 98

o Grammar: “and so recommend cultivars”

- Line 131

o The isolate numbers are mentioned but it is not clear what these represent. Where are these isolates from? Is this referring to a numbering system in Table S1?

- Line 133

o It is unclear what is meant by they were chosen “due to their considerable significance for improvement of the variety”.

- Line 133-137 and 137-138

o Repeated sentences

- Line 177

o Did you mean fixed linear regression model?

- Line 226, Fig 1

o Is there a particular reason/significance to using a sum of scores bar chart was utilized as opposed to visualizing the three race scores separately?

- Line 271

o Further analysis or explanation of Fig. 2d and its significance to your study would be beneficial.

- Line 316

o Should this be Fig.4?

- Line 344

o Could you further explain here how you determined the cultivars that were most likely sources of genetic resistance to ANT? How did Figure 5 help you draw these conclusions?

- Line 386

o Which chromosome is SNP ss715646017 on? It would be helpful to stay consistent and and add (Pv__) for all SNPs in this section.

- Line 412

o Is there any more information available about which disease resistance mechanisms these are?

6. PLOS authors have the option to publish the peer review history of their article (what does this mean?). If published, this will include your full peer review and any attached files.

Reviewer #1: No

Reviewer #2: **Yes: **Valerio Hoyos-Villegas

---

## [Author Response · Author response to Decision Letter 0]

8 Apr 2021

We are re-submitting the manuscript “Genome-wide association mapping reveals race-specific SNP markers associated with anthracnose resistance in carioca common beans” to be considered for publication as an original article in the PLOS ONE journal. We thank the Editor and the reviewers to taking the time to review our manuscript. All the suggestions were very important to the construction of our manuscript. We have accepted all the suggestions directly in the text, and response the doubts in this letter. We are available for any further clarification.

Reviewer 1

In general, the manuscript is poorly written with many grammatical errors and repetition of some phrases throughout the manuscript. The authors are strongly encouraged to re-write/thoroughly edit the manuscript. I have listed some specific examples below:

R: In advance, we would like to thank you for the thorough review carried out in our manuscript, which made it possible to improve the quality of the study. We agree with all your suggestions and accept any corrections made. The manuscript has been carefully revised by a native English speaker to improve the grammar and readability.

Abbreviations Define abbreviations upon first appearance in the text.

R: We reviewed the entire paper to ensure that the abbreviations were defined upon the first appearance in the text. (Lines 49, 53, 78, 105, 201, 210, 211, 389 and 494) 

L20 In the present study, using a carioca diversity panel (CDP) and a carioca segregating population (AM population), both genotyped using high-throughput genotyping technologies, we employ multiple interval mapping (MIM) and genome-wide association studies (GWAS) for resistance to the main ANT physiological races present in the country. Did you use MIM-Gene or any other software? The sentence is too long.

R: We did not use MIM-gene for linkage mapping. Indeed, we used the OneMap software (Margarido et al. 2007) as described by Almeida et al. 2021. We have divided the sentence. (Line 24)

L21 “study, using a carioca diversity panel (CDP, N=?) (state the number of lines in this panel) and a carioca segregating population (AM population)” Define abbreviations upon first appearance in the text (AM).

R: In a reference to the reviewer’s comment, we have added the number of lines and changed the abbreviation as suggested. (Line 22)

L22 “genotyped using high-throughput genotyping technologies,” Please name the genotyping technology used in your study.

R: We have included the name of the genotyping technology as requested. (Line 22 and 107)

L22 we employ multiple interval mapping (MIM) and genome-wide association studies (GWAS) for resistance to the main ANT physiological races present in the country. “Multiple interval mapping (MIM) and genome-wide association studies (GWAS) were used as mapping tools for the resistance genes to the major ANT…….

R: We have modified the sentence as suggested. (Line 24)

L24 In general, 14 SNPs “Define the abbreviations upon first appearance” and loci (Name the loci) associated with multiple races were also identified.

R: We appreciate the suggestion and performed the sentence alteration as requested. (Line 26)

L27 “known to be related to plant immunity were identified in the confidence interval (please clarify which confidence interval (CI) you are referring to, i.e., is it on Pv 04. Rephrase the sentence.

R: The sentence in this paragraph have been rewritten as requested. (Lines 27 – 29)

L27 Through discriminant analysis of principal components (DAPC), cultivar sources with high potential for durable anthracnose resistance were recommended. Do you mean that you used DAPC to identify cultivars with high anthracnose resistance?

R: We want to apologize to the reviewer as we really made a mistake in this part of the text. The DAPC was only possible to be applied and used for the identification of candidate cultivars due to the data used for the analysis. We added this information in the text. (Line 203-214)

L29 The MIM confirmed the presence of the Co-1 4 29 locus in the AND-277 30 cultivar “remove “but showed that the locus” which revealed that it was the only one associated with resistance to ANT race 65.

R: In response to the reviewer’s comment, we changed the sentence as suggested. (Line 33)

L31 This is the first study to “delete point out” and insert “identify” new resistance loci in the AND-277 cultivar.

R: We have altered the sentence as suggested. (Line 35)

L33 The new SNPs identified, especially those associated with more than one race, “remove -presented” present great potential for use in {Remove-the process of” marker-assisted (remove- selection) and early selection of inbred lines.

R: We have modified the text as suggested. (Line 38)

Introduction

L47 “the largest producer (remove “s “) and consumer (remove “s “) of beans in the world,

R: we performed the corrections of the sentence as suggested by the reviewer. (Line 51)

L49 The estimated production of “the last “list the year when this data was collected/reported) common bean crop in Brazil

R: We have changed the sentence adding the information requested. (Line 53)

L52 “of approximately 35% in relation to the varieties of the period)” which period? Be specific.

“the new variety quickly came to” Name the variety and rephrase the sentence.

R: We have modified the sentence as suggested, and we added the period to which it refers. (Line 56)

L53 “The (remove this-cultivars of the) carioca commercial group “add –cultivars” are characterized by cream-colored grain with brown stripes and high yield “add yields) (9) and (and “they “belong to the

R: We performed the correction of the sentence as suggested. (Line 58)

L67 “races 65, 73, and 81 as those of most frequent occurrence in 68 Brazil “replace with “as some of the most frequent occurring races in Brazil.

R: We have rephrased the sentence as the reviewer´s suggestion. (Line 72)

L69 The development of cultivars with “replace lasting, with durable” resistance ………..

R: We have replaced the term as suggested. (Line 74)

L70 “ensuring the “replace grower greater economic yield” with “higher yields” and less environmental contamination. (16).”

R: The sentence has been modified. (Line 75)

L73 Delete “Up to this time, (~ ) “Insert “About “ 25 loci with multiple alleles of…….” both of” replace with “from both” Andean and Mesoamerican origin

R: The sentence has been changed. (Line 78)

L75 “Remove “These” and replace with “The” loci of Mesoamerican origin “Delete “are” and insert “include” Co

R: The text has been modified as suggested. (Line 79)

L78 Pv11 “Add, respectively” (20,21,30–32,22–29). The “Co” loci “delete “of “and replace with “originating from “Andean origin.

R: The sentence has been reformulated as requested. (Line 83)

L88 reported many “remove “cultivar sources of “Add ANT resistance sources from mainly cultivars of Andean origin (40,42,45,46,52–55) “delete;”, and also some of Mesoamerican origin,

R: The sentence has been reformulated as requested. (Line 92)

L90 “delete” “they belonged” add “belonging to the black or special bean commercial classes.

R: We have altered the sentence as suggested. (Line 93)

L91 Replace “Obtain” with developing

R: The verb has been changed. (Line 94)

L92 Delete” For that reason”, “Delete tend to” i.e., Breeding programs routinely exploit only the elite germplasm “delete “of the variety”, which results in a narrow genetic base.

R: The sentence has been reformulated as requested. (Line 96)

L93 Delete “The difficulty is even greater when it is necessary to Investigate” and replace with “It is difficult to use Andean accessions as a source of resistance (why?). Delete “For that reason”, “delete the” identification of loci associated with ANT resistance in carioca bean is extremely important “delete “with the goal of achieving” add” in development/breeding of elite genotypes with resistance to multiple races.

R: In response to the reviewer’s comment, we changed the sentence, and we added the reason why it is difficult to use Andean accessions as a source of resistance to improving Mesoamerican cultivars. (Line 101)

L98 resistance and (delete so, add) recommendation of these cultivars, “delete, that are resistant to the main ANT races” in Brazil (delete- “alleles favorable” and “add”) will lead to genetic improvement of the common bean (delete crop).

R: The sentence has been reformulated as requested. (Line 104)

L99 Delete “For that purpose.”

R: We have removed the text from the manuscript as suggested. (Line 116)

L101 “Using high-throughput genotyping technologies”, name which genotyping technologies?

R: We totally agree with the reviewer and apologize for this mistake. The name of the genotyping technology used was added to the text. (Line 119)

L103 Why use the two parents? Was there any unique trait etc.?

R: We used that the AM population obtained from the crossing of both parents derived from different gene pools to maximize the genetic divergence and heterosis. Compared to allogamous species, heterosis in autogamous plants is rarely explored due to the high cost of hybrid seed production on a commercial scale. However, hybrid vigor must be assessed for the selection of superior genetic constitutions. Thus, the resulting AM population has large phenotypic divergence for several traits, and it was recently used in a study aimed at determining QTLs associated with angular leaf spot resistance (Almeida et al. 2021).

Materials and Methods

L108 Delete “the efforts of” add “genotypes from the main genetic breeding.

R: The sentence was altered according to the suggestion. (Line 113)

L109 The sentence is too long (L108-L111) Put a full-stop after 11 different institutions. These vary from the first…..

R: We have changed the sentence as suggested. (Line 114)

L127 The Am population was not is

R: We totally agree with the reviewer and performed the correction of the tense of the verb. (Line 133)

L140 Plot (delete was composed) consisted of one row with five plants

R: Indeed, the reviewer is totally correct. We changed the sentence as suggested. (Line 144)

L142 Delete of each, add per

R: In response to the reviewer’s comment, we changed the sentence as suggested. (Line 146)

L144 rephrase remove “remaining”, The seeds were transplanted and grown in the green house for nine days. Indicate the green house conditions i.e., humidity and temperature.

R: We appreciate the insightful comment and rephrased the sentence as suggested. However, the greenhouse where the plants stayed for nine days does not have temperature and humidity control. (Line 147)

L150 replace intensity with severity, replace “generated by the infection of the disease” with resulting from inoculation with ANT races”

R: We are grateful for this suggestion and altered the sentence in accordance. (Line 154)

L153 Delete cultivar (used as a resistant check)

R: We have modified the text according to the suggestion. (Line 157)

L154 Delete cultivar

R: We have modified the text according to the suggestion. (Line 158)

L156 Rephrase, Analysis of variance was used on all phenotypic data.

R: The sentence was rephrased as required. (Line 160)

L157 Delete, all and add, three races evaluated was done.

R: In response to the reviewer’s comment, we have changed the sentence as suggested. (Line 161)

L178 which software was used for CIM?

R: We used the One Map package (Margarido et al. 2007) to identify the QRL. We added this information in the text. (Line 180)

L182 Replace “Starting from with Using the QRL model identified in CIM, replace “sought” with identified

R: We have modified the sentence as suggested. (Line 187)

L184 What does conditional significance mean? Both models (complete and reduced) what does this mean/difference

R: For MIM mapping, the significance of the QRL identified by the CIM is defined by the comparison between the complete model (containing the QRL) and the reduced model (without the QRL). When the reduced model does not differ from the complete model, the QRL is not significant.

L207 Replace “best cultivars” with resistant cultivars vs susceptible cultivars (not worst)

R: In response to the reviewer’s comment, we have changed the sentence as suggested. (Line 2013)

L221 How did you measure the accuracy of the trials

R: The precision of the experiment was estimated through the selective accuracy (rgg) through the software Rbio. Selective accuracy is a parameter that refers to the correlation between the true genotypic value of the genetic treatment and the values estimated or predicted from the data obtained. (Line 161)

L294 The LOD score for each QRL is below the threshold of LOD = 3.0 and they seem to explain less than 10% of phenotypic variation indicating they are minor ones. How many environments and years was this replicated?

R: Indeed, the identified QRLs were of lesser effects. The experiments for phenotypic evaluation of anthracnose resistance were conducted in an extremely controlled environment (chamber), with constant temperature at 22° C, relative humidity above 95% and photoperiod of 12hrs. For guaranteed reasons, we choose to adopt a randomized block design with three replications. However, due to the high homogeneity of the inoculation chamber, no significance was observed for the block factor in the analysis of variance. 

L295 Name the C. lindemuthianum races used. The table should be stand alone/self explanatory

R: We added the C. lindemuthianum races used in the heading of the table so that it becomes self-explanatory. (Line 303)

L312 A total of 16 resistance candidate genes were identified for the range of both SNPs, with two encoding protein kinases and the remaining R genes. This is not clear, clarify.

R: We have modified the sentence in order to clarify it. (Line 322)

L346 which were most promising, delete “as sources of genetic resistance to ANT”. In addition to”,

R: We have performed the changes in the sentence as suggested. (Line 358)

L347 Beside being resistant to the three ANT races, “delete” “and having been grouped in the resistance cluster (i.e., favorable alleles)”, all of them are commercial …..

R: We have modified the text according to the suggestion. (Line 359)

L359 identification of genotypes “holding” rephrase

R: The sentence was rephrased as required. (Line 370)

L372 “programs tend to prefer to perform” replace with tend to do crosses among elite carioca cultivars

R: The sentence was rephrased as required. (Line 383)

L378 may be sustained by both data sets, which ones?

R: We agree that the meaning of the sentence was confusing. We believe that for parental selection towards anthracnose resistance, the best approach is to gather both phenotypic and genotypic information. The sentence has been rewritten. (Line 388)

 

Reviewer 2

General Comments

The introduction is very informative but leaves room for questions that are not addressed and/or clarified until the discussion. It may be beneficial to sufficiently address the topic of Andean vs. Mesoamerican gene pools in the introduction and then relate it to this particular study within the discussion section or hold off to address it in detail in the discussion session. Currently it is difficult to follow divided between the two sections.

R: In the first place, we would like to thank Professor Valerio Hoyos-Villegas for the correction and suggestions that made it possible to improve the quality of the manuscript. We agree with all your suggestions and corrections throughout the manuscript. We have added a brief information about the common bean gene pools in the introduction to facilitate understanding. (Lines 45-46)

Minor comments and suggestions

- First sentence and second sentence, line 16 - 18

Flip the order of the first and second sentence for clarity and re-organize wording of first sentence. Suggestion: “Brazil is the largest consumer of dry edible beans in the world, 70% of consumption is of the carioca variety. (Introduce the impacts of anthracnose on these bean crops, what does it do?) The most effective strategy to overcome anthracnose (ANT), a disease caused by the fungus Colletotrichum Lindemuthianum, is the development of resistant cultivars.”

R: We are incredibly grateful for the reviewer’s comment and accepted the suggestion. The beginning of the abstract really improved with the proposed changes. (Lines 16 – 20)

- Line 20 Omit “In the present study”

R: We have altered the sentence as suggested. (Line 22)

- Line 46 Omit “In this scenario”

R: We have changed the sentence as suggested. (Line 51)

- Line 49 It is unclear what sentence means, “estimated production of the last common bean crop in Brazil…”. Was this the harvest of a particular year?

R: In fact, the harvest was from the year 2020/2021. We added this information the sentence as suggested. (Line 53)

- Line 51 It is a bit unclear what is meant by the use of the word “superiority” in the context of this sentence.

R: We appreciate the comment and added additional information to the sentence for a better comprehension. (Line 56)

- Line 55 and 66 The abstract would benefit from adding the sentence, “Although the variety has high yield…”

R: We added the suggestion at the beginning of the abstract. (Line 17)

- Line 58 and 60 Redundant to include “high degree of pathogenicity” twice in back to back sentences

R: We performed the correction as suggested. (Line 63)

- Line 61 In scientific literature referring to anthracnose race identification, common wording I have seen is “susceptible” and “resistant” as opposed to “compatible” and “incompatible”. It may be good to stick to the re-occurring terminology.

(Perhaps also beneficial to refer to Pastor- Corrales here in this explanation of race identification)

R: We agree with the reviewer’s suggestion and performed changes in the text. (Line 65)

- Line 98 Grammar: “and so recommend cultivars”

R: In agreement to the reviewer’s correction, we have altered the expression. (Line 104)

- Line 131 The isolate numbers are mentioned but it is not clear what these represent. Where are these isolates from? Is this referring to a numbering system in Table S1?

R: In fact, the number refers to the code of these isolates in the IAC fungal library. We have added this information to the text. (Lines 138-139)

- Line 133 It is unclear what is meant by they were chosen “due to their considerable significance for improvement of the variety”.

R: We carried out the correction of the sentence, aiming to make it clearer that the isolates were selected based on the importance for the improvement of the carioca variety. Indeed, 65 and 81 are the most frequent occurring races in Brazil and 321 is one of the most virulent races. (Lines 140-141)

- Line 133-137 and 137-138 Repeated sentences

R: In response to the reviewer’s comment, we performed the correction of the text excluding the redundancies. 

 Line 177 Did you mean fixed linear regression model?

R: In fact, we meant that, and we performed the correction for clarifying the meaning. (Line 183)

- Line 226, Fig 1 Is there a particular reason/significance to using a sum of scores bar chart was utilized as opposed to visualizing the three race scores separately?

R: Truly speaking, the objective of using the sum of scores bar chart was to highlight and facilitate the visualization of cultivars resistant to all tested races.

- Line 316 Should this be Fig.4?

R: We appreciate the reviewer´s comment and performed the correction. (Line 326)

- Line 344 Could you further explain here how you determined the cultivars that were most likely sources of genetic resistance to ANT? How did Figure 5 help you draw these conclusions?

R: Definitely. Basically, we selected all cultivars that showed resistance for all ANT races. Then, we used only the markers associated with the QRL for the cluster analysis by DACP. For this analysis, we represented all alleles with a positive effect on resistance as "R" and alleles with a negative effect as "S". In addition, we added two control samples ("fake" genotypes), one containing only S alleles for all markers and the other only R alleles. In this way, the cultivars used for the analysis that were grouped with the control having resistance (only resistance alleles) are those with the greatest potential to be exploited as a source of resistance in further breeding programs. Besides the observed phenotypic resistance, these cultivars have the greatest number of favorable alleles. Figure 5 shows the clustering, with the cultivars in green, those that were grouped with the resistant control. 

- Line 386 Which chromosome is SNP ss715646017 on? It would be helpful to stay consistent and and add (Pv__) for all SNPs in this section.

R: SNP ss715646017 is on chromosome Pv07. We agree with the reviewer and add the chromosome information for all SNPs referred in this session of the manuscript. (Lines 320, 321, 325 and 340)

- Line 412 Is there any more information available about which disease resistance mechanisms these are?

R: Absolutely. Among the six genes in the interval in question, three are R genes (NB-LRR, nucleotide-binding, leucine-rich repeat) and the other are kinase-encoding ones. We complemented the information in the text. (Lines 424-426)

---

## [Decision Letter · Decision Letter 1]

3 May 2021

Genome-wide association mapping reveals race-specific SNP markers associated with anthracnose resistance in carioca common beans

PONE-D-21-03824R1

Dear Dr. Almeida,

We’re pleased to inform you that your manuscript has been judged scientifically suitable for publication and will be formally accepted for publication once it meets all outstanding technical requirements.

Kind regards,

Istvan Rajcan, Ph.D.

Academic Editor

PLOS ONE

Additional Editor Comments (optional):

Reviewers' comments:

Reviewer's Responses to Questions

**Comments to the Author**

1. If the authors have adequately addressed your comments raised in a previous round of review and you feel that this manuscript is now acceptable for publication, you may indicate that here to bypass the “Comments to the Author” section, enter your conflict of interest statement in the “Confidential to Editor” section, and submit your "Accept" recommendation.

Reviewer #2: All comments have been addressed

2. Is the manuscript technically sound, and do the data support the conclusions?

Reviewer #2: Yes

3. Has the statistical analysis been performed appropriately and rigorously? 

Reviewer #2: Yes

4. Have the authors made all data underlying the findings in their manuscript fully available?

Reviewer #2: Yes

5. Is the manuscript presented in an intelligible fashion and written in standard English?

Reviewer #2: Yes

6. Review Comments to the Author

Reviewer #2: (No Response)

7. PLOS authors have the option to publish the peer review history of their article (what does this mean?). If published, this will include your full peer review and any attached files.

Reviewer #2: **Yes: **Valerio Hoyos-Villegas

---

## [Editor Report · Acceptance letter]

5 May 2021

PONE-D-21-03824R1 

Genome-wide association mapping reveals race-specific SNP markers associated with anthracnose resistance in carioca common beans 

Dear Dr. Almeida:

I'm pleased to inform you that your manuscript has been deemed suitable for publication in PLOS ONE. Congratulations! Your manuscript is now with our production department. 

Kind regards, 

on behalf of

Dr. Istvan Rajcan 

Academic Editor

PLOS ONE